# REAL-TIME IMAGE DEMOIRÉING ON MOBILE DEVICES

**Yuxin Zhang[1] [\*], Mingbao Lin[1], Xunchao Li[1], Han Liu[2], Guozhi Wang[2], Fei Chao[1],**
**Shuai Ren[2], Yafei Wen[2], Xiaoxin Chen[2], Rongrong Ji[1,3,4] [†]**
[1]MAC Lab, School of Informatics, Xiamen University, [2]VIVO AI Lab,
[3]Institute of Artificial Intelligence, Xiamen University, [4]Pengcheng Lab

## ABSTRACT

Moiré patterns appear frequently when taking photos of digital screens, drastically degrading the image quality. Despite the advance of CNNs in image demoiréing, existing networks are with heavy design, causing redundant computation burden for mobile devices. In this paper, we launch the first study on accelerating demoiréing networks and propose a dynamic demoiréing acceleration method (DDA) towards a real-time deployment on mobile devices. Our stimulus stems from a simple-yet-universal fact that moiré patterns often unbalancedly distribute across an image. Consequently, excessive computation is wasted upon non-moiré areas. Therefore, we reallocate computation costs in proportion to the complexity of image patches. In order to achieve this aim, we measure the complexity of an image patch by designing a novel moiré prior that considers both colorfulness and frequency information of moiré patterns. Then, we restore image patches with higher-complexity using larger networks and the ones with lower-complexity are assigned with smaller networks to relieve the computation burden. At last, we train all networks in a parameter-shared supernet paradigm to avoid additional parameter burden. Extensive experiments on several benchmarks demonstrate the efficacy of our proposed DDA. In addition, the acceleration evaluated on the VIVO X80 Pro smartphone equipped with a chip of Snapdragon 8 Gen 1 shows that our method can drastically reduce the inference time, leading to a real-time image demoiréing on mobile devices. Source codes and models are released at `https://github.com/zyxxmu/DDA`.

## 1 INTRODUCTION

Moiré patterns (Sun et al., 2018; Yang et al., 2017b) describe an artifact of images that in particular appears in television and digital photography. In contemporary society, using mobile phones to take screen pictures has become one of the most productive ways to record information quickly. Nevertheless, moiré patterns occur frequently from the interference between the color filter array (CFA) of camera and high-frequency repetitive signal. The resulting stripes of different colors and frequencies on the captured photos drastically degrade the visual quality. Therefore, developing demoiréing algorithms has received long-time attention in the research community, and yet remains unsolved in particular when running algorithms on mobile devices.

Primitive studies on image demoiréing resort to traditional machine learning techniques such as low-rank and sparse matrix decomposition (Liu et al., 2015) and bandpass filters (Yang et al., 2017a). The rising of convolutional neural networks (CNNs) has vastly boosted the efficacy of image demoiréing (He et al., 2019; Zheng et al., 2020). However, the improved quantitative performance, such as PSNR (Peak Signal-to-Noise Ratio), comes at the increasing costs of energy and computation. For example, MBCNN (Zheng et al., 2020) eats up 4.22T floating-point operations (FLOPs) in order to restore a 1920×1080 smartphone-taken moiré image. Given that the moiré patterns mostly emerge in mobile photography, such massive computations carry considerable inference latency, preventing a real-time demoiréing experience from the users. Such a handicap could

---

[\*]Work done during Yuxin Zhang's internship at VIVO AI Lab
[†]Corresponding author: rrji@xmu.edu.cn

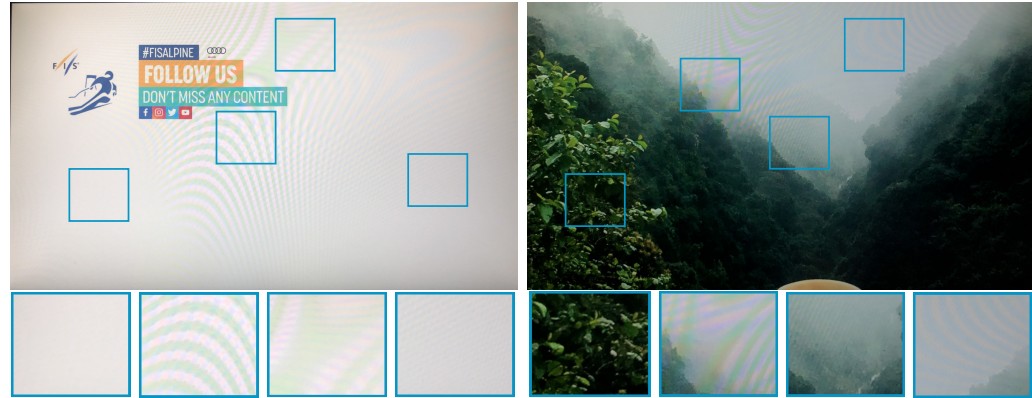

Figure 1: Images with moiré patterns. The blue boxes show a blue zoomed in area. Two phenomena can be observed: (1) The moiré complexity varies significantly across different areas of an image. (2) Moiré patterns are mainly characterized by frequency and colorfulness.

be more serious when it comes to video demoiréing. Therefore, it is of great need to bridge the technology gap between academia and industry.

To tackle the aforementioned issue, we initiate the first study on accelerating demoiréing networks towards a real-time deployment on mobile devices. Our motivation arises from empirical actuality in moiré images. As shown in Fig. 1, an image is often partially contaminated by the moiré pattern. Some areas are filled with intensive moiré stripes, some are much relieved while some are kept away from moiré pollution. It is natural to hand out computation to these moiré centralized areas but less to these diluted areas. In the extreme case, it is needless to cleanse uninfluenced areas. Unfortunately, existing methods (Sun et al., 2018; Zheng et al., 2020) have not distinguished the treatment to the different areas in an image. They not only waste excessive computation on non-moiré areas but also bring about side effects, such as over-whitened image contents. Therefore, reallocating computation costs in compliance with the complexity of a moiré area can be a potential solution to accomplish real-time image demoiréing on mobile devices.

Stimulated by the above analysis, we opt to split a whole image into several sub-image patches. To measure the patch moiré complexity, we introduce a novel moiré prior. As can be referred in Fig. 1, moiré patterns are featured with either high frequency or rich color information. Thus, we define the moiré prior as the product of frequency and color information in a patch. In detail, we model the frequency information by a Gaussian filter and the colorfulness metric is a linear combination of the mean and standard deviation of the pixel cloud in the RGB colour space (Hasler & Suesstrunk, 2003). Using this prior to measure the moiré complexity, each image patch is then processed by a unique network with its computation costs in proportion to the moiré complexity. In this fashion, larger networks are utilized to restore moiré centralized areas to ensure the recovery quality while smaller networks are leveraged to restore moiré diluted areas to relieve computation burden. Thus, we have a better tradeoff between the image quality and resource requirements on mobile devices.

Nevertheless, multiple networks lead to more parameter burden, which also causes deployment pressure due to the short-supply memory on mobile devices. To mitigate this issue, we leverage the supernet paradigm (Yang et al., 2021) to jointly train all networks in a parameter-shared manner. Concretely, we regard the vanilla demoiréing network as a supernet, and weight-shared subnets of different sizes are directly extracted from this supernet to process image patches of different demoiré complexity. During the training phase, each subnet is dynamically trained using corresponding image patches with similar moiré complexity. Consequently, the overall running overhead can be effectively reduced without introduction of any additional parameters.

We have conducted extensive experiments for accelerating existing demoiréing networks on the LCDMoiré (Yuan et al., 2019) and FHDMi (He et al., 2020) benchmarks. The results show that our dynamic demoiréing acceleration method, termed DDA, achieves an obvious FLOPs reduction even with PSNR and SSIM increases. For instance, DDA reduces 45.2% FLOPs of the state-of-the-art demoiréing network MBCNN (Zheng et al., 2020) with 0.35 dB PNSR increase. Furthermore, the

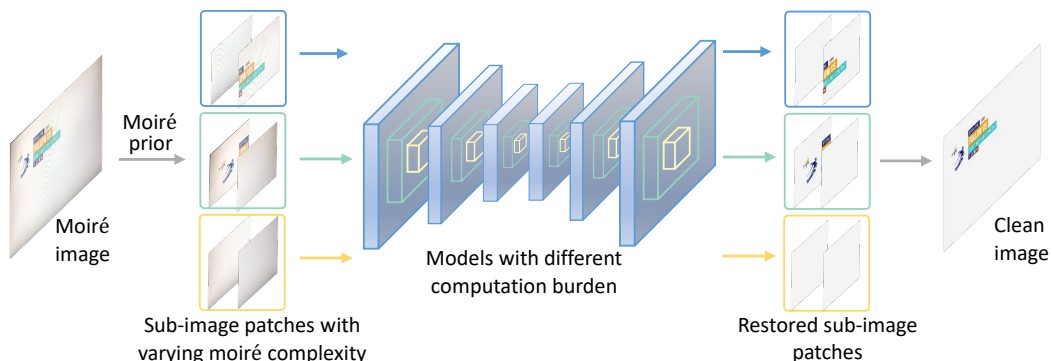

Figure 2: The framework of our proposed dynamic demoiréing acceleration method (DDA).

accelerated DMCNN (Sun et al., 2018) leads to real-time image demoiréing on the VIVO X80 Pro smartphone, with a tiny latency of 48.0ms when processing an image of 1920×1080 resolution.

This work addresses the problem of demoiréing network deployment on resource-limited mobile devices. The key contributions of this paper include: (1) A novel framework for accelerating image demoiréing networks in a dynamic manner. (2) An effective moiré prior to identify the demoiréing complexity of a given image patch. (3) Performance maintenance and apparent acceleration on modern smartphone devices.

## 2 RELATED WORK

### 2.1 IMAGE DEMOIRÉING

Image demoiréing aims at removing moiré patterns on captured images. Early work mainly focuses on manually designed algorithms with the aid of low-rank & sparse matrix decomposition (Yang et al., 2017a; Liu et al., 2015). With the explosion and popularity of deep learning, extensive demoiréing networks are proposed in recent years to achieve moiré removal in an end-to-end manner (Sun et al., 2018; He et al., 2019). As a pioneering work, (Sun et al., 2018) proposed a multi-scale network structure to remove moiré patterns at different frequencies and scales. (He et al., 2019) dived into designing specific learning schemes to resolve the unique properties of moiré patterns including frequency distributions, edge information and appearance attributes. (Zheng et al., 2020) reformulated the image demoiréing problem as moiré texture removal and color restoration, and proposed MBCNN (Zheng et al., 2020) which consists of a learnable bandpass filter to learn the frequency prior and a two-step tone mapping mechanism to restore color information. FHDe2Net (He et al., 2020) uses a global branch to eradicate multi-scale moiré patterns and a local branch to reserve fine details. (Liu et al., 2020) further proposed WavleNet to handle demoiréing in the wavelet domain. The same dilemma for the aforementioned networks is their huge computation burden, which greatly prohibits the practical deployment on mobile devices.

### 2.2 DYNAMIC NETWORKS AND SUPERNETS

Dynamic networks adapt the network structures or parameters *w.r.t.* different inputs (Kong et al., 2021; Huang et al., 2017; Bolukbasi et al., 2017). Due to the advantages in accuracy performance and computation efficiency, dynamic networks have received increasing research interest in recent years. A comprehensive overview of dynamic networks can be found at (Han et al., 2021). Supernets are a type of dynamic network that reserves weight-shared sub-networks of multiple sizes within only one network, and randomly samples these sub-networks for training (Chen et al., 2022; Yang et al., 2021; Yu & Huang, 2019). According to the constraints of available resources, different sub-networks with varying widths and resolutions can be adaptively chosen during the testing phase without introducing additional parameter burden. Inspired by these studies, our proposed DDA involves the supernet paradigm by dynamically allocating image patches with different demoiréing complexity to their corresponding sub-networks.

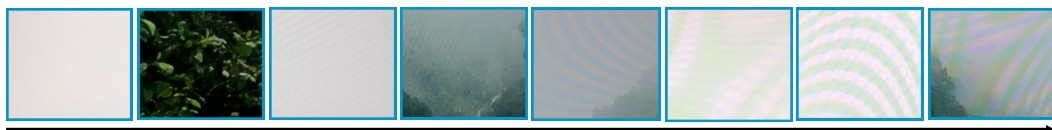

Moiré Complexity Score

Figure 3: The sorted sub-image patches *w.r.t.* moiré complexity scored by our proposed moiré prior.

## 3 METHODOLOGY

Fig. 2 manifests the general framework of our dynamic demoiréing acceleration method (DDA). A moiré image is firstly split into several sub-image patches, which are then reorganized into different groups in conformity with the patch moiré complexity. This is achieved by a novel moiré prior that considers both color and frequency information of moiré patterns. We detail it in Sec. 3.1. For the purpose of real-time image demoiréing on mobile devices, we train multiple networks of different complexity to process patches in different groups. In the training and testing phases, image patches with higher-complexity are fed to larger networks and patches with lower-complexity are dealt with smaller networks. Sec. 3.2 gives the implementations. Finally, as an alternative for training separate networks that result in more parameters, we regard each one as a subnet of the vanilla demoiréing network (supernet), leading to a weight-shared training paradigm as depicted in Sec. 3.3.

### 3.1 MOIRÉ PRIOR

The complexity degree of moiré patterns can be determined by a human, however, it is lavish and laborious to manually define the complexity for all patches in every moiré image. Many former works on dynamic networks (Han et al., 2021; Kong et al., 2021) train an additional module to adapt the network *w.r.t.* different inputs, which, however, brings unexpected parameters and computations for its compositions of several convolutional or fully-connected layers. Given our plan of deploying demoiréing networks on mobile devices, such a solution is not feasible, or at least not optimal.

We propose a novel moiré prior to measure the moiré complexity of an image in a fast manner. The motivation for this prior comes from an in-depth observation on moiré images. As can be inferred from Fig. 3, moiré patterns vary a lot in frequency and colorfulness. Customarily, a perceptible moiré pattern is highlighted by either high frequency or rich color information. Therefore, a prior reflecting both image frequency and colorfulness can be an efficacious method to model the intensity of moiré patterns. Denoting a moiré image as $X$, we first decompose it into sub-image patches as $\{x_i\}_{i=1}^N$. For a specific patch $x$, we use the Gaussian high-pass filter (Dogra & Bhalla, 2014) with a standard deviation of 5 for the Gaussian distribution to extract the frequency information as $\mathcal{F}(x)$. To measure the patch colorfulness, we consider a linear combination of the mean and standard deviation of the pixel cloud in the RGB colour space (Hasler & Suesstrunk, 2003):

$$C(x) = \sqrt{\sigma^2(x_R - x_G) + \sigma^2(0.5(x_R + x_G) - x_B)}$$
$$+0.3\sqrt{\mu^2(x_R - x_G) + \mu^2(0.5(x_R + x_G) - x_B)}, \tag{1}$$

where $\mu(\cdot)$ and $\sigma(\cdot)$ are the mean and standard deviation functions, $x_R$, $x_G$, $x_B$ denote the R, G, B color channels, respectively. Here $0.3$ is a parameter found by (Hasler & Suesstrunk, 2003) through maximizing the correlation between the experimental data and the colorfulness metric. Refer to (Hasler & Suesstrunk, 2003) for more principles of measuring image colorfulness. Therefore, our proposed moiré complexity score using frequency and colorfulness priors is finally defined as:

$$M(x) = C(x) \cdot \mu\big(F(x)\big), \tag{2}$$

where $\mu(\cdot)$ is the mean function. Fig. 3 shows that $M(x)$ can be a reliable metric for evaluating the patch moiré complexity. Notice that, without building any extra network module, the operations of our moiré prior become highly cheap, bringing negligible computation burden.

### 3.2 DYNAMIC DEMOIRÉING ACCELERATION

In image demoiréing, a moiré-polluted image $X$ is expected to restore to moiré-free ground-truth in natural scenes. A traditional demoiréing process is formulated using CNNs as:

$$Y = \mathcal{F}(X; \Theta), \tag{3}$$

where $\mathcal{F}$ standards for the demoiréing network with its parameters denoted as $\Theta$. As can be seen from Eq. (3), existing methods restore all areas of a moiré image with the same network $\mathcal{F}$, which wastes excessive computation since the moiré complexity varies significantly across different areas of an image as aforementioned in Sec. 1. This violates our goal of real-time image demoiréing on mobile devices. A natural way for demoiréing acceleration is to reallocate computation costs according to the complexity of a moiré area.

To that effect, we reorganize the sub-image patches $\{x_i\}_{i=1}^{N}$ of $X$ in an ascending order of their moiré complexity score defined in Eq. (2), and then split the ordered patches into $M$ groups denoted as $\{G_1, G_2, ..., G_M\}$. Each group $G_i$ contains these image patches, moiré complexity scores of which range from the top-$\left((i-1)\cdot\lceil\frac{N}{M}\rceil+1\right)$ to -$\left(i\cdot\lceil\frac{N}{M}\rceil\right)$ smallest among all. Then, we construct $M$ different demoiréing networks $\{\mathcal{F}_i\}_{i=1}^{M}$ with parameters of different sizes as $\{\Theta_i\}_{i=1}^{M}$, and process each image patch $x \in G_i$ using the $i$-th network $\mathcal{F}_i$ as:

$$y = \mathcal{F}_i(x; \Theta_i | x \in G_i). \tag{4}$$

In our setting, the complexity of $\mathcal{F}_i$ is smaller than that of $\mathcal{F}_{i+1}$ such that smaller-complex image patches can be handled by networks with low computation costs, and vice versa. Eq. (4) dynamically accelerates the derivation of Eq. (3) by assigning computation costs in line with the degree of moiré complexity. Meanwhile, the recovery quality is still ensured as moiré centralized areas are restored using larger networks. Finally, the moiré-free output of our dynamic demoiréing acceleration method (DDA) is obtained by concatenating patch outputs of all networks:

$$Y = concat\big(\mathcal{F}_1(x; \Theta_1 | x \in G_1), \mathcal{F}_2(x; \Theta_2 | x \in G_2), ..., \mathcal{F}_M(x; \Theta_M | x \in G_M)\big), \tag{5}$$

where $concat()$ concatenates the output patches to construct a moiré-free full image.

### 3.3 SUPERNET TRAINING

Though the aforesaid procedure benefits reduction of the overall computation costs, the challenge arises in respect of parameter burden when deploying our demoiréing method on mobile devices featured with short-supply memories. For a simple case, setting the largest network $\mathcal{F}_M$ as the vanilla demoiréing network $\mathcal{F}$, additional parameters of $\sum_{i=1}^{M-1}(|\Theta_i|)$ are introduced in total.

To solve this, in place of training networks $\{F_i\}_{i=1}^{M}$ in isolation, we further propose to use the supernet paradigm (Yang et al., 2021) to train and infer all networks in a parameter-shared manner. In detail, the vanilla demoiréing network $\mathcal{F}$ is regarded as a supernet and its parameters $\Theta$ are partly shared by $\mathcal{F}_i$. Supposing the network width of $\mathcal{F}_i$ is $W_i$, we inherit the first $W_i$ proportion of convolution filter weights to the subnet $\mathcal{F}_i$, denoted as $\Theta[W_i]$. Consequently, the network $\mathcal{F}_i$ becomes a subnet of $\mathcal{F}$. Therefore, our moiré-free output is finally reformulated as:

$$Y = concat\big(\mathcal{F}(x; \Theta[W_1]|x \in G_1), \mathcal{F}(x_2; \Theta[W_2]|x \in G_2), ..., \mathcal{F}(x; \Theta[W_M])|x \in G_M\big). \tag{6}$$

As a consequence, image demoiréing can be effectively accelerated even without any additional parameter burden, which finally reaches our target for a real-time deployment on mobile devices.

## 4 EXPERIMENT

### 4.1 EXPERIMENT SETUP

#### 4.1.1 DATASETS

There are three main public datasets for image demoiréing. (1) LCDMoiré dataset (Yuan et al., 2019) from the AIM19 image demoiréing challenge consists of 10,200 synthetically generated image pairs including 10,000 training images, 100 validation images and 100 testing images at $1024{\times}1024$ resolution. (2) FHDMi dataset (He et al., 2020) contains 9,981 image pairs for training and 2,019 for testing with $1920{\times}1080$ resolution. (3) The TIP2018 dataset (Huang et al., 2017) consists of real photographs constructed by photographing images with $400{\times}400$ resolution from ImageNet (Deng et al., 2009) displayed on computer screens. In this paper, we conduct experiments on the LCDMoiré and FHDMi datasets. We do not consider the TIP2018 benchmark since the resolution is too small to meet our target for image demoiréing on mobile devices from which the captured images generally have an extremely high resolution with $1920{\times}1080$ or higher.

Table 1: Ablation study for applying different width list configurations in supernet training.

| W | FHDMi | | | AIM | | |
|---|---|---|---|---|---|---|
| | PSNR | SSIM | Params | PSNR | SSIM | Params |
| $\{0.4, 0.5, 0.6\}$ | 22.81 | 0.8124 | 10.61M | 41.68 | 0.9869 | 10.61M |
| $\{0.25, 0.5, 0.75\}$ | 23.07 | 0.8766 | 11.88M | 41.43 | 0.9852 | 11.88M |
| $\{0.1, 0.5, 0.9\}$ | 21.03 | 0.7988 | 13.15M | 40.21 | 0.9791 | 13.15M |

#### 4.1.2 EVALUATION PROTOCOLS

We adopt the widely-used metrics of PSNR (Peak Signal-to-Noise Ratio) and SSIM (Structure Similarity) to conduct a quantitative comparison for demoiréing performance. We also report the color difference between restored images and the clean images using the CIE DeltaE 2000 (Sharma et al., 2005) measurement, denoted as $\Delta E$. The Float Points Operations (denoted as FLOPs) and network latency per image on VIVO X80 Pro smartphone with the Snapdragon 8 Gen 1 chip of networks are reported as the accelerating evaluation.

#### 4.1.3 BASELINES

We choose to accelerate DMCNN (Sun et al., 2018) and MBCNN (Zheng et al., 2020) to verify the efficacy of our DDA method. DMCNN is a pioneering network for image demoiréing with a multi-scale structure. MBCNN is a state-of-the-art demoiréing network, which consists of a learnable bandpass filter to learn the frequency prior and a two-step tone mapping mechanism to restore color information. We report the results of baseline models as well as their compact versions by slimming the network width based on our re-implementation on the PyTorch framework (Paszke et al., 2019).

#### 4.1.4 IMPLEMENT DETAILS

Our implementation of DDA is based on the PyTorch framework (Paszke et al., 2019), with the group number $M = 3$, width list $W = \{0.25, 0.5, 0.75\}$ on FHDMi and $W = \{0.4, 0.5, 0.6\}$ on LCDMoiré. We split the original images of LCDMoiré and FHDMi datasets into sub-image patches of 512×512 and 640×540, respectively. Then, we classify the sub-image patches into multiple groups with different moiré complexity using our proposed moiré prior. We train the supernet using Adam (Kingma & Ba, 2014) optimizer. The initial learning rate and batch size are set to 1e-4 and 4 in all experiments. During training, we iteratively extract a batch of image pairs within a specific class of moiré complexity, which are used to train the subnet of corresponding width extracted from the supernet. For DMCNN, we give 200 epochs for training with the learning rate divided by 10 at the 100-th epoch and 150-th epoch. For MBCNN, we follow (Zheng et al., 2020) to reduce the learning rate by half if the decrease in the validation loss is lower than 0.001 dB for four consecutive epochs and stop training once the learning rate becomes lower than 1e-6. All experiments are run on NVIDIA Tesla V100 GPUs.

### 4.2 PERFORMANCE ANALYSIS

In this section, we perform detailed performance analysis on the different components of our DDA including the supernet training paradigm and moiré prior.

#### 4.2.1 SUPERNET TRAINING

We first conduct experiments to investigate how the hyper-parameters in the supernet training influence the performance of DDA, *w.r.t*, the width list configuration $W$. The experiments are conducted on the FHDMi (He et al., 2020) and LCDMoiré (Yuan et al., 2019) datasets using MBCNN. We can observe from Tab. 1 that a dispersed configuration $W = \{0.25, 0.5, 0.75\}$ performs better on the FHDMi dataset, while a compact configuration $W = \{0.4, 0.5, 0.6\}$ works better on the LCDMoiré dataset. To explain, LCDMoiré is a synthetic dataset, where the moiré patterns distribute more balanced compared with FHDMi that is captured using embedded cameras. Generally speaking, a more discrete width list guarantees our purpose of dynamically removing moiré patches of different complexity in real scenarios.

Table 2: Ablation study for dynamic acceleration with/without supernet training.

| Method | PSNR | Params |
|---|---|---|
| w supernet | 23.62 | 11.88M |
| w/o supernet | 23.15 | 30.16M |

Table 3: Ablation study for restoring image patches from the same group with different widths.

| Network | $G_1$ | $G_2$ | $G_3$ |
|---|---|---|---|
| MBCNN-0.75× | 24.03 | 22.21 | 21.29 |
| MBCNN-0.5× | 24.12 | 22.02 | 20.22 |
| MBCNN-0.25× | 23.99 | 21.81 | 19.69 |

Table 4: Ablation study for the moiré prior.

| Prior | PSNR |
|---|---|
| C | 22.44 |
| F | 22.18 |
| C+F | 22.42 |
| Ours | 23.62 |

Table 5: Quantitative results for demoiréing acceleration on the FHDMi dataset.

| Method | PSNR | SSIM | $\Delta E$ | FLOPs | FLOPs↓ | Params | Params↓ | Latency |
|---|---|---|---|---|---|---|---|---|
| DMCNN | 21.69 | 0.7731 | 6.67 | 699.16G | 0.0% | 1.43M | 0.0% | 69.4ms |
| DMCNN-0.75× | 21.56 | 0.7704 | 6.81 | 476.62G | 31.4% | 0.99M | 29.9% | 55.2ms |
| DMCNN-0.5× | 21.11 | 0.7691 | 7.14 | 314.82G | 54.8% | 0.68M | 52.1% | 47.9ms |
| DMCNN-0.25× | 20.63 | 0.7655 | 7.98 | **208.73G** | **70.1%** | **0.48M** | **66.6%** | **41.8ms** |
| DDA | **21.86** | **0.7708** | **6.55** | 333.39G | 52.3% | 0.99M | 29.9% | 48.0ms |
| MBCNN | 23.27 | 0.8201 | 5.38 | 4.22T | 0.0% | 14.21M | 0.0% | 259.8ms |
| MBCNN-0.75× | 22.51 | 0.8113 | 6.11 | 3.05T | 27.3% | 11.88M | 16.4% | 192.4ms |
| MBCNN-0.5× | 22.12 | 0.8077 | 6.32 | 2.07T | 47.5% | 9.92M | 30.2% | 147.2ms |
| MBCNN-0.25× | 21.83 | 0.7991 | 6.54 | **1.28T** | **60.8%** | **8.36M** | **41.2%** | **119.7ms** |
| DDA | **23.62** | **0.8293** | **5.21** | 2.13T | 45.2% | 11.88M | 16.4% | 147.1ms |

Besides, Tab. 2 compares the performance for accelerating MBCNN on LCDMoiré between supernet training and respectively training each sub-network. It can be seen that supernet training does not lead to performance degradation and it drastically reduces the parameter burden compared with simultaneously keeping multiple sub-networks. The result well demonstrates the effectiveness of our DDA for practical deployment.

### 4.2.2 MOIRÉ PRIOR

We further analyze the performance of the proposed moiré prior. The experiments are conducted on the FHDMi dataset using MBCNN. We use networks of different widths to infer different groups of pictures with different complexity classified by our proposed moiré prior. The results for Tab. 3 show that all widths perform similarly for the easiest group, while larger width significantly outperforms smaller width for the group with highest moiré complexity. Such results demonstrate the effectiveness of our proposed moiré prior operator, and also validate our point that using large networks to restore patches with low moiré complexity wastes massive computation resources.

At last, we investigate three variants of our proposed moiré prior including (1) only using high-frequency information (denoted as F), (2) only using colorfulness information (denoted as C), (3) adding two information scores instead of multiplication in Eq. (2) (denoted as C+F). As shown in Tab. 4, all variants result in worse performance, which well demonstrates the efficacy of our proposed moiré prior that considers both color and frequency properties of moiré patterns. It is worth mentioning that C+F implies domination of the colorfulness measurement due to the fact that scores given by C are generally two orders of magnitude larger than those of F in our observation. As a result, C+F leads to a similar performance to C. In contrast, by multiplying both scores, our prior offers reliable moiré complexity for a given image patch.

### 4.3 QUANTITATIVE COMPARISON

Tab. 5 and Tab. 6 report the quantitative results of our DDA for accelerating DMCNN and MBCNN. On FHDMi, DDA surprisingly improves the PSNR of MBCNN by 0.35 dB even with a FLOPs reduction of 45.2%. We attribute such results to that the original baseline assigns the same network to restore the areas with very few or no moiré patterns, which may damage the original details of the image. Consequently, the poor performance barricades the usage of demoiréing networks in practical deployment. In contrast, DDA leverages the smallest network to restore these non-moiré areas, leading to a better global demoiréing effect. Besides, we demonstrate the effectiveness of DDA by comparing MBCNN accelerated by it with several state-of-the-art demoiréing networks including MDDM (Cheng et al., 2019), MopNet (He et al., 2019), FHDe[2]Net (He et al., 2020) on FHDMi dataset. As can be seen from Tab. 7, DDA can outperform other networks regarding both complexity reduction and demoiréing performance. For instance, DDA surpasses FHDe[2]Net by 0.69

Table 6: Quantitative results for demoiréing acceleration on the LCDMoiré dataset.

| Method | PSNR | SSIM | $\Delta E$ | FLOPs | FLOPs $\downarrow$ | Params | Params $\downarrow$ | Latency |
|---|---|---|---|---|---|---|---|---|
| DMCNN | **34.58** | **0.9612** | 1.76 | 353.11G | 0.0% | 1.43M | 0.0% | 35.2ms |
| DMCNN-0.75$\times$ | 33.41 | 0.9589 | 1.84 | 242.24G | 31.4% | 0.99M | 29.9% | 28.0ms |
| DMCNN-0.5$\times$ | 32.99 | 0.9604 | 1.90 | 159.67G | 54.8% | 0.68M | 52.1% | 24.2ms |
| DMCNN-0.25$\times$ | 31.75 | 0.9547 | 2.27 | **105.42G** | **70.1%** | **0.48M** | **66.6%** | **21.1ms** |
| DDA | 34.19 | 0.9601 | **1.73** | 158.71G | 55.1% | 0.78M | 45.4% | 24.4ms |
| MBCNN | **43.95** | **0.9909** | **0.69** | 2.14T | 0.0% | 14.21M | 0.0% | 132.1ms |
| MBCNN-0.75$\times$ | 41.67 | 0.9853 | 0.90 | 1.55T | 27.3% | 11.88M | 16.4% | 98.1ms |
| MBCNN-0.5$\times$ | 41.31 | 0.9844 | 0.94 | 1.05T | 47.5% | 9.92M | 30.2% | 76.3ms |
| MBCNN-0.25$\times$ | 40.78 | 0.9801 | 0.94 | **0.65T** | **60.8%** | **8.36M** | **41.2%** | **61.2ms** |
| DDA | 41.68 | 0.9869 | 0.85 | 1.09T | 46.9% | 10.61M | 25.4% | 75.2ms |

Table 7: Performance comparison between MBCNN accelerated by DDA and state-of-the-art demoiréing networks on the FHDMi dataset.

| Method | DMCNN | MDDM | MopNet | FHDe$^2$Net | MBCNN | MBCNN-DDA |
|---|---|---|---|---|---|---|
| PSNR | 21.69 | 20.83 | 22.76 | 22.93 | 23.14 | **23.62** |
| SSIM | 0.7731 | 0.7343 | 0.7958 | 0.7885 | 0.8201 | **0.8293** |
| FLOPs | **0.41T** | 0.97T | 6.26T | 11.41T | 4.22T | 2.13T |
| Params | **2.37M** | 8.01M | 12.40M | 13.57M | 14.21M | 10.61M |

Table 8: Performance comparison between MBCNN accelerated by DDA and state-of-the-art demoiréing networks on the LCDMoiré dataset.

| Method | DMCNN | MDDM | MDDM+ | MopNet | MBCNN | MBCNN-DDA |
|---|---|---|---|---|---|---|
| PSNR | 34.58 | 42.49 | 43.44 | 42.02 | **43.95** | 41.68 |
| SSIM | 0.9612 | 0.9940 | **0.9960** | 0.9872 | 0.9909 | 0.9869 |
| FLOPs | 476.62G | **472.38G** | 440.44G | 3.16T | 2.14T | 1.09T |
| Params | **2.37M** | 8.01M | 6.55M | 12.40M | 14.21M | 10.61M |

dB PSNR with even far fewer FLOPs (2.13T for DDA and 11.41T for FHDe$^2$Net), which shows the correctness and effectiveness of our perspective for reallocating computation costs in proportion to the moiré complexity of image patches.

As to LCDMoiré, compared with DMCNN-0.75 which simply infers the whole image, our DDA dynamically assigns computation resources with respect to moiré complexity of patches, retaining a better PSNR performance (34.19 dB for DDA and 33.41 dB for DMCNN-0.75) and more FLOPs reduction (55.1% for DDA and 31.4% for DMCNN-0.75). Meanwhile, DDA achieves a noticeable latency reduction of 56.9ms for accelerating MBCNN (75.2ms for DDA and 132.1ms for the baseline), enabling a real-time image demoiréing on mobile devices. Nevertheless, a noticeable performance drop of PSNR is still observed (41.68 dB for DDA and 43.95 dB for the full model), and comparison results with state-of-the-art networks including MDDM (Cheng et al., 2019), MDDM+ (Cheng et al., 2021) and MopNet (He et al., 2019) in Tab. 8 also suggest a relatively poor result of DDA than its performance on FHDMi. Here we argue that the LCDMoiré dataset is built on simulating the aliasing between CFA and the screen's LCD subpixel, which results in images with different distributions of moiré patterns compared with smartphone-captured moiré images. Compared with smartphone-captured FHDMi with different moiré distributions, the moiré patterns in LCDMoiré are much more uniform and their cropped moiré patches are of similar complexity. This explains the relatively poor performance of DDA on the LCDMoiré dataset. Note that our approach even improves the performance of baseline models with less computation burden on FHDMi and achieves superior performance in comparison with SOTA networks. Given our motivation for the practical deployment of image demoiréing in real cases, the efficacy of the proposed method is still affirmative.

## 4.4 QUALITATIVE COMPARISON

In addition to the quantitative results, Fig. 4 further displays the visualization results of restored images on the FHDMi dataset. Results on the LCDemoiré dataset can be found in Appendix A.2. As can be observed, uniformly performing the same accelerating rate for the whole image (MBCNN-

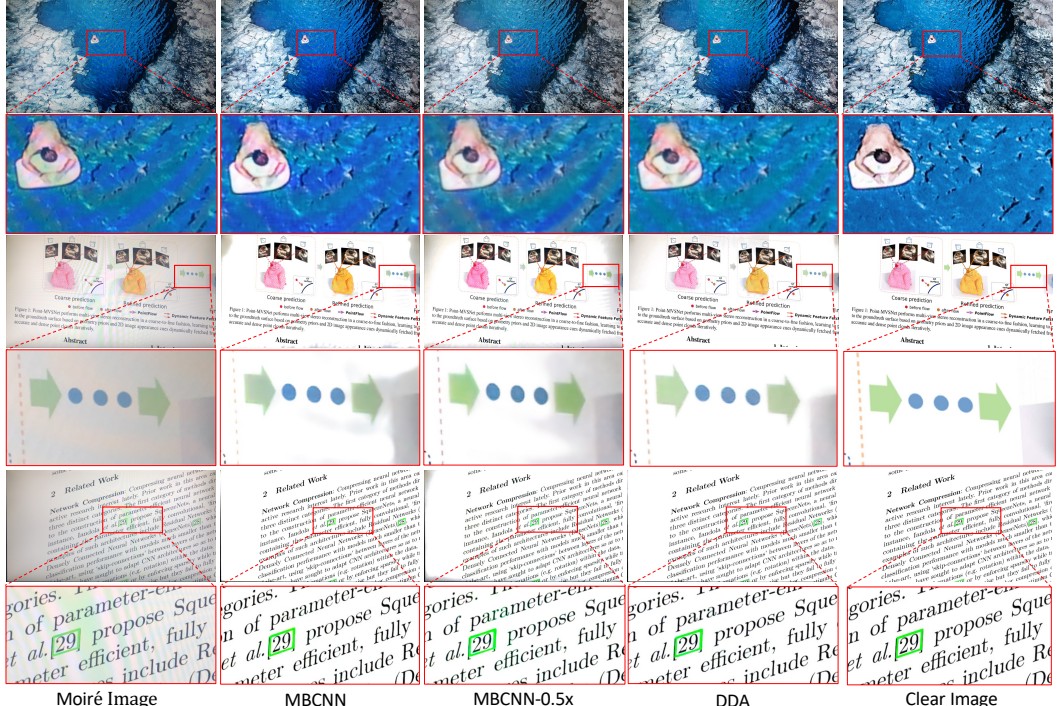

| Moiré Image | MBCNN | MBCNN-0.5x | DDA | Clear Image |

Figure 4: Visual quality comparison for accelerating MBCNN on the FHDMi dataset. The red boxes show a zoomed in area for better observation.

0.5×) drastically degrades the performance as the areas with dense moiré patterns do not receive enough computation resources to be efficiently restored. In contrast, by dynamically allocating the computational resources, our DDA achieves promising demoiréing quality compared with the original network. The efficacy of our proposed DDA for accelerating demoiréing networks for practical application is therefore well demonstrated.

## 5 LIMITATION AND FUTURE WORK

We further discuss the limitations of our DDA, which will be our future focus. Firstly, DDA simply divides all images into equal-number patches in each class, laying some avenues for future research in devising image-aware classification priors. Besides, our limited computing facilities prevent us from accelerating other demoiréing networks with varying structures (Liu et al., 2020; He et al., 2019). More validations are expected to be performed to further demonstrate the efficacy of DDA.

## 6 CONCLUSION

In this paper, we have presented a novel dynamic demoiréing acceleration method (DDA) to reduce the huge computational burden of existing networks towards real-time demoiréing on mobile devices. Our DDA is based on the observation that the moiré complexity is highly unbalanced across different areas of an image. On this basis, we propose to split the whole image into sub-patches, which are then regrouped according to their moiré complexities measured by a novel moiré prior that considers both the frequency and colorfulness information. Then, we use models with different sizes to restore patches in each group. In particular, larger networks are utilized to restore moiré centralized areas to ensure the recovery quality while smaller networks are leveraged to restore moiré diluted areas to relieve computation burden. To avoid the additional parameter burden caused by retaining multiple networks, we further leverage the supernet paradigm to jointly train the networks in a parameter-shared manner. Results on several benchmarks demonstrate that our method can effectively reduce the computation costs of existing networks with negligible performance degradation, enabling a real-time demoiréing on current smartphones.

## ACKNOWLEDGEMENT

This work is supported by the National Science Fund for Distinguished Young (No.62025603), the National Natural Science Foundation of China (No.62025603, No. U1705262, No. 62072386, No. 62072387, No. 62072389, No, 62002305, No.61772443, No. 61802324 and No. 61702136) and Guangdong Basic and Applied Basic Research Foundation (No.2019B1515120049).

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

# A APPENDIX

## A.1 MORE VISUALIZATION RESULTS FOR THE MOIRÉ PRIOR

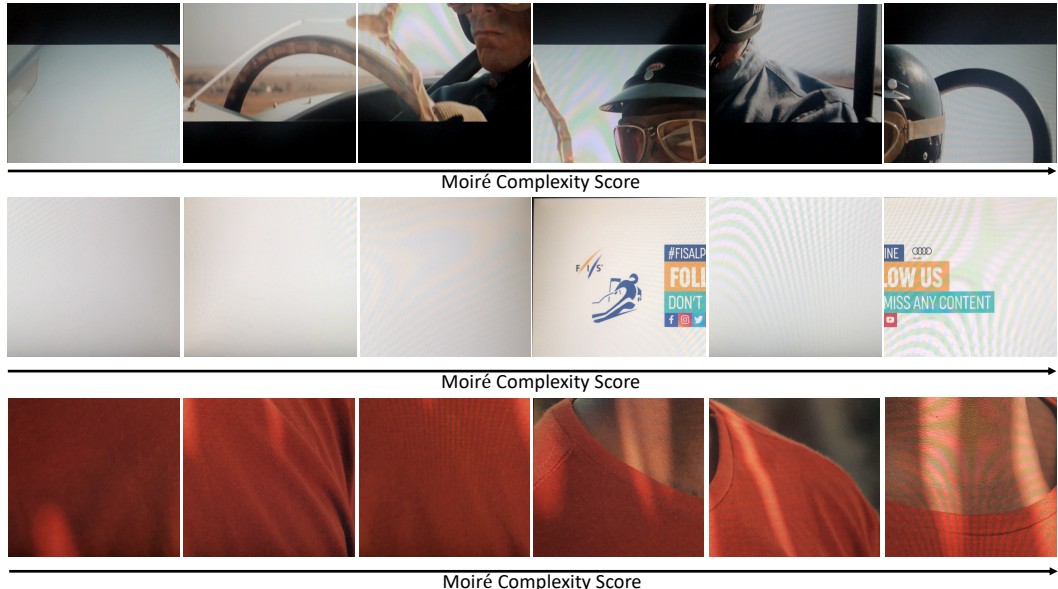

Figure 5: The sorted sub-image patches cropped from the FHDMi dataset according to moiré complexity scores given by our proposed moiré prior.

## A.2 QUALITATIVE RESULTS ON THE LCDMOIRÉ DATASET

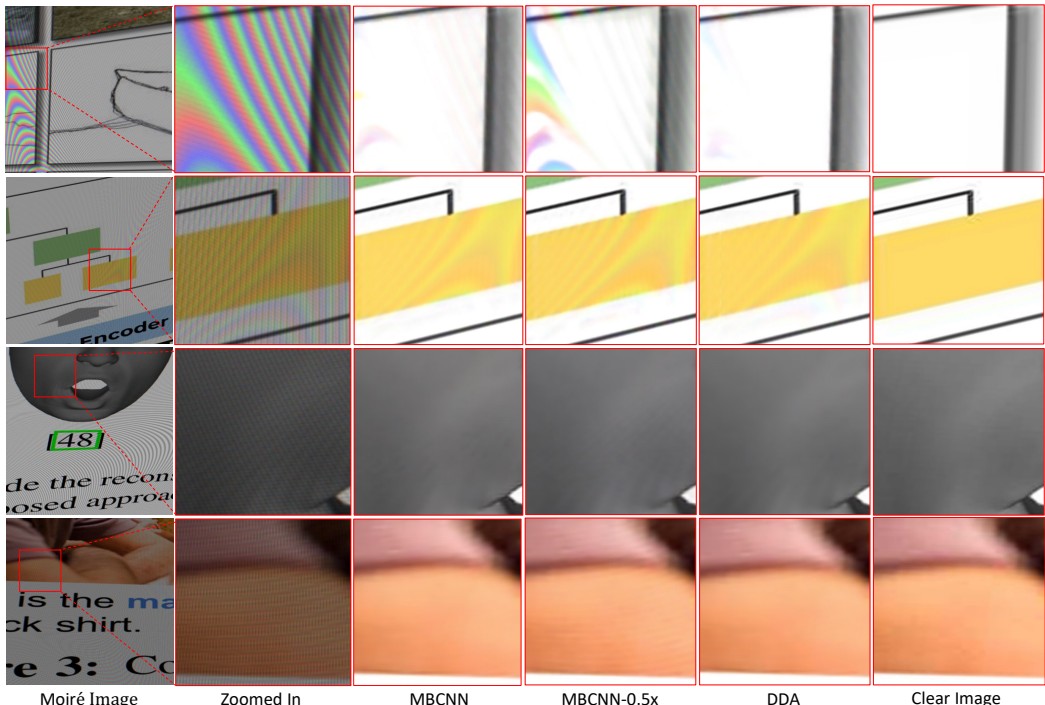

Figure 6: Visual quality comparison for accelerating MBCNN on the LCDMoiré dataset. The red boxes show a zoomed in area for a better observation.

Table 9: Training time comparison on the FHDMi and LCDMoiré datasets. We report NVIDIA Tesla V100 GPU days.

| Dataset | DMCNN | DMCNN-DDA | MBCNN | MBCNN-DDA |
|---------|-------|-----------|-------|-----------|
| LCDMoiré | 0.42 | 0.61 | 3.89 | 5.77 |
| FHDMi | 2.04 | 3.11 | 8.19 | 10.02 |

## A.3 TRAINING TIME COMPARISON

In this section, we report the training time of MBCNN (Zheng et al., 2020), DMCNN (Sun et al., 2018) and their accelerated version by DDA. The results in Tab. 9 suggest heavier training consumption of DDA compared with the vanilla demoiréing networks. The additional training time stems from more training iterations per epoch in our supernet since the original datasets have been split into multiple patches. Nevertheless, we stress our goal in this paper is to perform a real-time deployment with its predominant advantage at the inference efficiency.

