# OpenReview forum: "Real-Time Image Demoir$\acute{e}$ing on Mobile Devices"
_ICLR.cc/2023/Conference — ICLR 2023 poster_

### Official Review · Reviewer_LYcM · 2022-10-23

**Confidence:** 4
**Correctness:** 3
**Technical Novelty And Significance:** 2
**Empirical Novelty And Significance:** 2
**Recommendation:** 5

**Clarity, Quality, Novelty And Reproducibility:**

The paper’s clarity could be improved in places.  Some of these points may relate to things not discussed in the paper and impact reproducibility:

1.	Some patches will have no moire patterns.  How does the supernet deal with these?  Perhaps there could be a “passthrough” option for patches that have no moire, they require no processing at all by the demoire network.

2.	The network architecture of DDA isn’t so clear.  How does the supernet split the computation for demoire?  It appears the supernet is also applied to MBCNN, but it wasn’t clear to this reviewer how this is achieved, as MBCNN is a multi-scale architecture and not so easy to split into different computational blocks.  Perhaps the author rebuttal can comment on this.

3.	What are the parameters used to compute Equation 2?  For example, the parameters of the Gaussian high pass kernel?  Without describing these it because difficult to reproduce the paper.

4.	It wasn’t clear if the source code and pretrained models will be released should the paper be accepted.

5.	After Equation 5, the paper states that output patches are concatenated to reconstruct the full image.  In this reviewer’s experience, doing this type of processing may result in artifacts at patch boundaries, because each patch was processed independently by the network.  Are such artifacts present in the proposed results?  How to handle these artifacts?

6.	In Table 2, MBCNN performs best on LCDMoire in terms of PSNR and SSIM.  It would be best to make it bold in this table.  Usually a bold font is used to denote the best performing method.


The main originality comes from the supernet approach and how patches are characterized.  The supernet appears in other literature.  The characterization of patches is interesting and has some novelty, although the components in the score are known.  The paper has some novel elements but may read more as a systems paper bringing together known concepts into a practical working algorithm.

Smaller issues:

•	Throughout the paper, please use “complexity” when discussing computation.  For example, in the abstract, please change “higher-complex” to “higher-complexity”; and “lower-complex” to “lower-complexity”.  Please note this issue appears several times in the paper, not just the abstract.

•	Page 1, the phrase “on offline mobile devices” seems unusual.  In what sense are the mobile devices offline?  Note most smartphones have accelerators capable of running neural networks.  Perhaps the word “offline” can be removed.

•	Page 1, please change “contaminated by the moire.” to “contaminated by the moire pattern.”

•	Page 2, please change “Detailedly,” to “In detail,”

•	Page 2, please change “resource-hungry” to “resource-limited”

•	Page 3, please change “pioneer work” to “pioneering work”.  Note this issue also appears on Page 6.

•	Page 4, please change “by human” to “by a human”

•	Page 4, please change “or full-connected” to “or fully-connected”

•	Page 4, please change “not the optimal” to “not optimal”

•	Throughout the paper, please change “zooming-in” to “zoomed in”

•	Table 2 uses the term LCDMoire dataset, but Table 4 calls it AIM.  Please choose one for consistency.

•	Page 9, please change “nowadays smartphones” to “current smartphones”


**Strength And Weaknesses:**

Demoireing of images is an interesting and hard problem, recently receiving attention in the computer vision community particularly through the use of deep learning techniques.  As argued by the paper, it is common to take photographs of screens using a smartphone, but the resulting image may be corrupted by moire patterns.  They can occur anytime high frequency patterns interfere with the camara color filter array.  This paper addresses the challenge of real-time demoire as existing approaches have focused primarily on demoire performance and not speed.  The method adopts the supernet strategy to dynamically process patches based on their moire complexity.  The more complex patches receive more computation compared to simpler ones.  The moire complexity is estimated by combining frequency and color estimators together into a score applied to patches.  The score is novel although its components are fairly well established.  The paper shows improvements in speed are possible using the supernet approach.

Strengths:

1.	The paper is the first to address real-time moire.  The paper doesn’t really describe why real-time performance is needed however.  Conceivable for photographs, if it takes up to 500 ms to demoire an image most users won’t notice as there is a lag between taking the photo and reviewing the captured photo.  Perhaps the real-time moire is important for video processing.  Perhaps the paper could articulate why real-time more is important, although it seems likely that real-time demoire has more use cases than a slower demoire approach.

2.	The method to characterise the patches in terms of their moire complexity is interesting.  The ranking in Figure 2 is compelling, with the severity of the moire pattern increasing from left to right.  The examples in Figure 2 seem to be mostly grayscale however; it would be more convincing if they were more colorful as color is one of the measurements in the scoring criteria.


3.	The paper appears to be the first to apply a supernet approach to demoire.  This provides a practical approach to reduce computational complexity to perform fast demoire, by applying the computation where it is most needed.

Weaknesses:

1.	The paper seems to overstate the problem given the relative improvement produced by the proposed technique.  For example, in the abstract, the paper describes existing networks “causing massive computation burden”.  In Table 1, DMCNN requires 69.4ms latency whereas DDA requires 48.0ms.  DDA is certainly an improvement, but if DMCNN is a massive computation burden one might wonder if DDA also could be described similar as it is only 31% faster.  Perhaps the language could be toned down.

2.	Page 2, the paper claims that existing methods bring about side effect, such as over-whitened image contents.  However, the results don’t seem to show this, and indeed MBCNN seems to have better PSNR on LCDMoire than the proposed method.  It would be more convincing if this claim was substantiated somehow, for example in the results section.

3.	The paper has novelty in the way it scores patches in terms of their moire complexity and then uses a supernet architecture, however it relies on components derived from other papers.

4.	Some clarity issues raise hold the paper back.  Clarity is discussed in the next section.


**Summary Of The Paper:**

This paper presents an approach for real-time demoire on mobile devices.  Demoireing is an interesting (and hard) problem of removing moire patterns from images, which can easily appear due to high frequency patterns in the scene interfering with the camera color filter array.  It is hard because the moire patterns span a large set of frequencies and colors.  The paper proposes to divide the image into patches, and for each patch assess its moire complexity.  The complexity of the demoire network is adaptive based on the moire complexity; with easier to handle patches receiving less computation compared to harder ones.  This way, the approach can provide the computation where it’s needed, saving unnecessary computations on easy patches.  This is implemented using a supernet approach, and shown to save computation.  Experimental results show improved PSNR and runtime on the FHDMi dataset, and slightly lower PSNR but improved runtime on the LCDMoire dataset compared to MBCNN.  However the method outperforms DMCNN on both datasets.

**Summary Of The Review:**

In summary, this paper is addressing an important and difficult problem of demoire.  The supernet approach, coupled with the way to find moire-contaminated patches is interesting and has some (limited) novelty.  The faster results are appreciated by this reviewer, and real-time performance may have important use cases, however these are not articulated in the paper.  The paper seems overly critical of previous work given its results underperform MBCNN on one dataset and are 31% faster than DMCNN on another dataset.  Nonetheless, the paper reflects an advance on the problem on fast demoire, and the paper is the first to explicitly address this problem.  The paper is held back due to clarity issues, which impact reproducibility.

---

> ### Author Response · Authors · 2022-11-11
> **Reply to Reviewer LYcM: Part I**
>
> Thank you for your very detailed review and constructive comments. We address your comments and questions in the following.
>
> **Q1**: The paper doesn’t really describe why real-time performance is needed however.  Conceivable for photographs, if it takes up to 500 ms to demoir$\acute{e}$ an image most users won’t notice as there is a lag between taking the photo and reviewing the captured photo.  Perhaps the real-time moir$\acute{e}$ is important for video processing.  Perhaps the paper could articulate why real-time more is important, although it seems likely that real-time demoir$\acute{e}$ has more use cases than a slower demoir$\acute{e}$ approach.
>
> **A1**: Thank you for such an insightful comment. We first point out that demoir$\acute{e}$ing does not necessarily happen at the photographing stage. When shooting landscapes or portraits, moir$\acute{e}$ patterns rarely appear. Therefore, some of mobile devices do not automatically run the demoir$\acute{e}$ing algorithm once photos are taken. Instead, it leaves an option icon for users to decide whether to post-process the potential moir$\acute{e}$. In this case, a real-time experience is much more important. In addition, as you said, the benefits of real-time demoir$\acute{e}$ing will be multiplied when it comes to the video demoir$\acute{e}$ing captured by smartphones. We have included this description in Sec.1 of our revised manuscript to make our motivation more convincing.
>
> **Q2**: The method to characterise the patches in terms of their moir$\acute{e}$ complexity is interesting.  The ranking in Figure 2 is compelling, with the severity of the moir$\acute{e}$ pattern increasing from left to right.  The examples in Figure 2 seem to be mostly grayscale however; it would be more convincing if they were more colorful as color is one of the measurements in the scoring criteria.
>
> **A2**: Many thanks for this insightful suggestion. Now we have added more visualization results of colorful moir$\acute{e}$ images in Appendix A.1.
>
> **Q3**: The paper seems to overstate the problem given the relative improvement produced by the proposed technique.  For example, in the abstract, the paper describes existing networks “causing massive computation burden”.  In Table 1, DMCNN requires 69.4ms latency whereas DDA requires 48.0ms.  DDA is certainly an improvement, but if DMCNN is a massive computation burden one might wonder if DDA also could be described similar as it is only 31\% faster.  Perhaps the language could be toned down.
>
> **A3**: We appreciate your careful review to point out this and we have modified the word ”massive” to “redundant” to make our language usage more rigorous.
>
> **Q4**: Page 2, the paper claims that existing methods bring about side effect, such as over-whitened image contents.  However, the results don’t seem to show this, and indeed MBCNN seems to have better PSNR on LCDMoir$\acute{e}$ than the proposed method.  It would be more convincing if this claim was substantiated somehow, for example in the results section.
>
> **A4**: Thanks for this insightful comment.  Here we argue that LCDMoir$\acute{e}$ dataset is built on simulating the aliasing between CFA and the screen’s LCD subpixel. Compared with smartphone-captured FHDMi with different moir$\acute{e}$ distributions, the moir$\acute{e}$ patterns in LCDMoir$\acute{e}$ are much more uniform and their cropped moir$\acute{e}$ patches are of similar complexity. This explains the relatively poor performance of our approach on LCDMoir$\acute{e}$ dataset. Please note that our approach even improves the performance of baseline models with less computation burden on FHDMi and achieves superior performance in comparison with SOTA networks. Given our motivation for the practical deployment of image demoir$\acute{e}$ing in real cases, we believe the efficacy of the proposed method is still affirmative. We have added the above discussion in Sec.4.3 of our revised manuscript.

---

> ### Author Response · Authors · 2022-11-11
> **Reply to Reviewer LYcM: Part II**
>
> **Q5**: The paper has novelty in the way it scores patches in terms of their moir$\acute{e}$ complexity and then uses a supernet architecture, however it relies on components derived from other papers.
>
> **A5**: Thanks for this comment. We admit that the supernet architecture is derived from other papers, but we would like to highlight the novelty of DDA for the first exploration in accelerating demoir$\acute{e}$ing networks towards a real-time deployment on mobile devices along with a novel moir$\acute{e}$ prior, which is highly appreciated by all other reviewers. Thus, we believe our work can be a considerable contribution to both academia and industries.
>
> **Q6**: Some patches will have no moir$\acute{e}$ patterns. How does the supernet deal with these?  Perhaps there could be a “passthrough” option for patches that have no moir$\acute{e}$, they require no processing at all by the demoir$\acute{e}$ network.
>
> **A6**: This is a very legitimate question and we hope this answer will clarify things and lift what made you suspicious.  As formulated in Eq. (5), our supernet allocates the subnets with the least computation costs (0.25$\times$ costs compared with the vanilla supernet in our practice) to restore the patches with the smallest moir$\acute{e}$ complexity scores. Self-confidently, a patch with no moir$\acute{e}$ patterns can be directly dealt with a ``passthrough'' option as you mentioned, but we can not precisely determine whether a patch contains moir$\acute{e}$ patterns or not. If we always choose to directly pass the patches with the smallest moir$\acute{e}$ complexity scores, several patches with diluted moir$\acute{e}$ patterns cannot be restored as well. Therefore, our choice is to handle these patches with a relatively smaller subnet in order to guarantee the demoir$\acute{e}$ing performance. Nevertheless, this issue you brought to us definitely offers a new trail for further reducing the computation cost as a future work.
>
> **Q7**: The network architecture of DDA isn’t so clear.  How does the supernet split the computation for demoir$\acute{e}$?  It appears the supernet is also applied to MBCNN, but it wasn’t clear to this reviewer how this is achieved, as MBCNN is a multi-scale architecture and not so easy to split into different computational blocks.  Perhaps the author rebuttal can comment on this.
>
> **A7**:  As described in Section 3.3, the supernet extracts different proportions of convolution filter weights with respect to a given width list to generate subnets with varying computation costs. Indeed, there is no gap for generalizing the supernet mechanism to multi-scale networks like MBCNN as the weights-extraction operation happens inside specific convolution layers, instead of the whole network blocks/stages.
>
>
> **Q8**: What are the parameters used to compute Equation 2?  For example, the parameters of the Gaussian high pass kernel?  Without describing these it because difficult to reproduce the paper.
>
> **A8**: Thanks for this constructive question. We use the Gaussian high-pass filter with a standard deviation of $5$ for the Gaussian distribution to extract high-frequency information, which has been added to Sec.3.1 of our revised paper. Meanwhile, the description of the colorfulness measurement has also been enriched according to the suggestion of Reviewer Wo5f.
>
> **Q9**: It wasn’t clear if the source code and pretrained models will be released should the paper be accepted.
>
> **A9**: Actually, we had already released the source code and pretrained models in the supplementary material. Now this have been commented in the abstract of the revised manuscript.

---

> ### Author Response · Authors · 2022-11-11
> **Reply to Reviewer LYcM: Part III**
>
> **Q10**: After Equation 5, the paper states that output patches are concatenated to reconstruct the full image.  In this reviewer’s experience, doing this type of processing may result in artifacts at patch boundaries, because each patch was processed independently by the network.  Are such artifacts present in the proposed results?  How to handle these artifacts?
>
> **A10**: You are right. Such artifacts infrequently happens on images with high-colorfulness at the crop boundary. In practice, this can be eliminated by cropping patches with a few overlap pixels and concatenating patches by averaging these overlapping pixels. We would also like to clarify that such a crop-concat operation is also neccessary for traditional demoir$\acute{e}$ing methods as the limited peak-calculation for smartphones can not afford to handle a whole image with high-resolution. Thus, such boundary artifacts cannot be concluded as a disadvantage for our method.
>
> **Q11**: In Table 2, MBCNN performs best on LCDMoir$\acute{e}$ in terms of PSNR and SSIM.  It would be best to make it bold in this table.  Usually a bold font is used to denote the best performing method.
>
> **A11**: Thank you very much for bringing this up. We have fixed this issue in the revised manuscript.
>
> **Q12**: Small issues.
>
> **A12**: Thanks for bringing these issues to our attention. They have been fixed now.
>
> Thank you once again for the time and efforts that have been devoted to our paper.  Overall, we believe our work addresses an important gap between industries and academia for a real-time image demoir$\acute{e}$ing on mobile devices. Note again that our code and models are available in the supplementary material and we have made a major effort in our revised version to reduce the clarity issues, which we hope can successfully resolve your major concern for the reproducibility of our work. If such is the case, we would greatly appreciate it if you would consider increasing the score.

---

> ### Author Response · Authors · 2022-11-17
> **Reminder**
>
> Dear reviewer LYcM,
>
> Thanks again for your valuable time and insightful comments. As the deadline for the Author/Reviewer discussion is approaching, it would be nice of you to let us know whether our answers have solved your concerns so that we can better improve our work. We are happy to provide any additional clarifications that you may need.
>
> Best regards!
>
> Paper 254 Authors

---

> > ### Comment · Reviewer_LYcM · 2022-11-17
> > **Thank you for the rebuttal and revision**
> >
> > Thank you for a carefully prepared rebuttal and revision, which clarifies the points that were uncertain earlier.  Overall this reviewer thinks this paper has value and has raised the overall assessment.  At the same time, the paper is an amalgamation of other methods applied to demoire, so the reviewer was hoping for more novelty for ICLR and therefore recommends marginally below acceptance.  However, the other reviewers are more enthusiastic about this paper -- if the paper were accepted to the conference that would be ok for me.

---

### Official Review · Reviewer_Wo5f · 2022-10-23

**Confidence:** 5
**Correctness:** 3
**Technical Novelty And Significance:** 3
**Empirical Novelty And Significance:** 3
**Recommendation:** 8

**Clarity, Quality, Novelty And Reproducibility:**

- The paper is very clear in most of the parts. As stated before, my only complain is that I believe section 4.3.1 should be put before, in a way that allows presenting some supernet results already in Tables 1,2,3.

- I am satisfied by the quality of the work.

- The novelty is good. Even if it seems quite a straighforward idea; it was not proposed before, and that is actually a very positivie thing.

- The authors seems quite straigforward to reproduce, although I would strongly encourage the authors to release their code -there are not comment on that in the manuscript-.



**Details Of Ethics Concerns:**

I am quite against that the authors have just put in blod their results in Tables 1, and 2. This leads to the impression that their PSNR and SSIM is always better, as it is standard in the community to mark in bold face the results for the best methods.

**Strength And Weaknesses:**

Strenghts:
- The idea is interesting, and addresses a real problem in current mobile devices. Dividing the patches according the level of Moiré is clearly and advantage, even for obtaining better results -as shown in Tables 1 and 2, because it avoids the over-whitening of the clear parts of the images.

- As already mentioned, the results are state-of-the-art or very close to it, with a reduction of FLOPs of around 50%.

- Also the paper is mostly well written (although I would put section 4.3.1 just after 4.1.4).

Weaknesses:
- The authors write about the color part of the Moiré prior (Eq.1) in a way that shows clearly that they have not put enough attention on understanding the color part. They just select one equation from another paper and give a light -to not say worng- explanation of it. Let me detail this. Authors say that the used a measure in the CIELAB color space... but the equation they present is in RGB color space. I agree when studying the equation, that it is definitelu inspired in an opponent color space -from which CIELAB is an example-, because of the relations applied between the R,G, and B channels... However, the CIELAB color space is a non-linear transform from RGB, and in the equation presented here, there is not a non-linearity! I strongly recommend the authors to rewrite this part, stating that they apply a Colorfulness formula on the RGB color space, that is inspired in the chromatic -because luminance is clearly ignored- part of color opponent spaces.

- Regarding to the previous comment (Eq. 1), even if the formula comes from another paper, authors need to briefly explain why there is a relation of 0.3 to 1 in the standard deviation and the mean.

-  In Eq.3, authors need to clarify who is $\mu$. Is it the mean after computing the high-pass filter? Also, if that is the case, I am surprised about how well both measures combine... as there is a direct multiplication of both. Are they therefore on the same order of magnitude? That should be stated. If not, one may lead to thing that if one is much more larger than the other, will be clearly dominating the prior, and that will mean than the other term is actually meaningless.

- Finally, as the paper are basing their Moiré Prior in some color measure, I believe they should at least add one color metric to their results. I recommend CIE DeltaE.

**Summary Of The Paper:**

This paper presents a image demoireing method that focus on its use on mobile phones. This is important as current methods are extremely costly, and cannot therefore be embedded in current devices. The authors propose to first, divide the image into patches, second, given a proposed Moire prior, classify the patches according to the level of Moire that is presented in them, and third, process ech of this patches depending on the Moire level. In orer to make this efficient and to not fight with the parameter burden; the authors propose to have a supernet -in particular a Vanilla Net-, which allows the share of weights for the subnetworks.

**Summary Of The Review:**

The paper tackles an improtant problem in current mobile phones, and present an idea that works. Also, it is mostly well written. The only downside is the explanation from the Color Equations, which clearly show that the authors are not from the color community, This said, I believe the paper can be accepted, given that the authors properly correct the explanation about the color parts as explained in the weaknesses section, and add a color metric to the manuscript.

---

> ### Author Response · Authors · 2022-11-11
> **Reply to Reviewer Wo5f**
>
> We sincerely appreciate your careful review, positive feedback, and constructive comments.  Especially, your comments around the moir$\acute{e}$ prior helps us improve that section quite a lot. Please kindly see our responses to your comments below.
>
> **Q1**: I strongly recommend the authors to rewrite this part, stating that they apply a Colorfulness formula on the RGB color space.
>
> **A1**: We appreciate your professional suggestion and apologize for our inadequate understanding of the color space. We have corrected the description in our revised manuscript as your suggested (see Sec.1 and Sec.3.1).
>
> **Q2**: Regarding to the previous comment (Eq.1), even if the formula comes from another paper, authors need to briefly explain why there is a relation of 0.3 to 1 in the standard deviation and the mean.
>
> **A2**: Thank you for the valuable feedback. We have added relevant explanations in Sec.3.1 of our updated manuscript.
>
> **Q3**: In Eq.3, authors need to clarify who is $\mu$. Is it the mean after computing the high-pass filter? Also, if that is the case, I am surprised about how well both measures combine... as there is a direct multiplication of both. Are they therefore on the same order of magnitude? That should be stated. If not, one may lead to thing that if one is much more larger than the other, will be clearly dominating the prior, and that will mean than the other term is actually meaningless.
>
> **A3**: We apologize that our original description is not clear enough and causes confusion.  You are right that $\mu$ represents the mean function and we directly multiply the scores given by colorfulness and high-frequency metrics. Nonetheless, they are not in the same order of magnitude and the scores given by the colorfulness metric are two orders of magnitude larger than the high-frequency metric. This is why we choose to multiply these two scores instead of adding them, which actually leads to your concern that the colorfulness metric will dominate the high-frequency metric. In contrast, multiplication ensures sufficient participation for both metrics. To further demonstrate our point, we have conducted an ablation study for adding these two scores in Sec.4.2.2 and the results suggest that adding the scores leads to a similar performance with merely using the colorfulness metric, while they both lag behind our proposed moir$\acute{e}$ prior. The aforementioned clarifications and ablation studies have been added to our revised manuscript (see Sec.3.1 and Sec.4.2.2 for details).
>
> **Q4**: Finally, as the paper are basing their Moiré Prior in some color measure, I believe they should at least add one color metric to their results. I recommend CIE DeltaE.
>
> **A4**: Thanks. The comparison results for the CIE DeltaE 2000 have been added to our revised manuscript (see Tab.5 and Tab.6).
>
> **Q5**: Section 4.3.1 should be put before, in a way that allows presenting some supernet results already in Tables 1,2,3
>
> **A5**: We have accordingly reorganized the section order in the revised paper as suggested.
>
> **Q6**: I would strongly encourage the authors to release their code there are not comment on that in the manuscript.
>
> **A6**: Indeed, we had already released the code and models in the supplementary material, which has been stressed in the revised manuscript.
>
> **Q7**: I am quite against that the authors have just put in blod their results in Tables 1, and 2. This leads to the impression that their PSNR and SSIM is always better, as it is standard in the community to mark in bold face the results for the best methods.
>
> **A7**: Thank you for this notice and sorry for the mistake here. We have now fixed it in the revised paper.

---

> > ### Comment · Reviewer_Wo5f · 2022-11-15
> > **Read the answer**
> >
> > I have read the answer provided by the authors. I keep my ranking of 8: Accept, good paper.

---

### Official Review · Reviewer_9cpB · 2022-10-24

**Confidence:** 4
**Correctness:** 3
**Technical Novelty And Significance:** 3
**Empirical Novelty And Significance:** 2
**Recommendation:** 6

**Clarity, Quality, Novelty And Reproducibility:**

- Clarity: The writing is easy to follow. The method is clearly discussed, except the use cases of previous works such as supernet training for which it is not clear how the parameters were shared or how much time would it consume to train in such a manner compared to individual small CNN models or a full image CNN.

- Quality: There are some concerns about the quality of the reconstructions as discussed in W2, W3, and W4 above.

- Novelty: The proposed approach utilizes previous methods for the shared training and achieving components of the moire complexity metric. However, it could still be considered novel for the application in demoireing.

**Strength And Weaknesses:**

- S1: Proposing a method to sort image patches in terms of their moire complexity.

- S2: Proposing a method to reduce the memory requirement of image demoireing while the drop in image quality of the reconstructed output is less than just simply reducing the network width.

- W1: The numbers in bold within the tables presenting quantitative results in tables 1,2, and 3 are not representing best results but just the proposed method's results. This could be misleading for the readers.

- W2: The comparison to SOTA methods is only provided for the FHDMi dataset and the same comparison for LCDMOIRE dataset is missing.

- W3: The MBCNN method has higher quantitative performance numbers for the LCDMOIRE dataset [2]. For example, it achieves a PSNR of 45.08 in table 6 of that paper. What is the reason for this performance reduction?

- W4: Some missing methods form SOTA comparisons that has achieved better quantitative results on the LCDMOIRE dataset include Islab-zju with PSNR of 44.07 as shown in table 1 of [1], MDDM with PSNR of 42.49 as shown in table 1 of [3], and MDDM+ with PSNR of 43.44 as shown in table 1 of [4]. It is also not clear without direct comparison if the proposed method achieves a better memory-quality trade off compared to the mentioned methods.

[1] Yuan S, Timofte R, Slabaugh G, Leonardis A, Zheng B, Ye X, Tian X, Chen Y, Cheng X, Fu Z, Yang J. Aim 2019 challenge on image demoireing: Methods and results. In International Conference on Computer Vision Workshop (ICCVW) 2019.

[2] Zheng B, Yuan S, Yan C, Tian X, Zhang J, Sun Y, Liu L, Leonardis A, Slabaugh G. Learning frequency domain priors for image demoireing. IEEE Transactions on Pattern Analysis and Machine Intelligence. 2021.

[3] Cheng X, Fu Z, Yang J. Multi-scale dynamic feature encoding network for image demoiréing. In International Conference on Computer Vision Workshop (ICCVW) 2019.

[4] Cheng X, Fu Z, Yang J. Improved multi-scale dynamic feature encoding network for image demoiréing. Pattern Recognition. 2021 ;116:107970.

**Summary Of The Paper:**

The paper attempts to propose a solution to clean moire which includes patterns of colored stripes with varied frequencies captured by a camera degrading the visual quality of images. Instead of using a large CNN to clean a whole input image, it targets image patches with lighter CNNs. In order to determine the width of the CNN to be used for each patch, a complexity score based on frequency and colorfulness is utilized. Also, to reduce the parameter burden of utilizing multiple CNNs at the time of deployment, a shared weight training scheme called supernet training is utilized. The resulting solution provides a trade-off for image quality and runtime latency, outperforming compared methods and enabling higher quality real-time mobile device deployment.

**Summary Of The Review:**

A method to train a model for image demoireing that can be utilized with reduced run-time latency at the time of deployment is provided. I have some concerns about missing comparisons to the SOTA methods, both lacking a dataset and some well performing methods. I would not the paper an improvement and ready for acceptance before knowing the results of such comparisons. Besides, a few clarifications on the training process could improve the paper.

---

> ### Author Response · Authors · 2022-11-11
> **Reply to Reviewer 9cpB**
>
> Thank you very much for your constructive comments. We address them below.
>
> **Q1**: The numbers in bold within the tables presenting quantitative results in tables 1,2, and 3 are not representing best results but just the proposed method's results. This could be misleading for the readers.
>
> **A1**: Sorry for our misleading and we have corrected them in our revised manuscript.
>
> **Q2**: The comparison to SOTA methods is only provided for the FHDMi dataset and the same comparison for LCDMOIRE dataset is missing.  Some missing methods from SOTA comparisons that has achieved better quantitative results on the LCDMOIRE dataset. It is also not clear without direct comparison if the proposed method achieves a better memory-quality trade off compared to the mentioned methods.
>
> **A2**: We further perform experiments and the comparison on LCDMoir$\acute{e}$ dataset has been added to Tab. 8 of our revised manuscript. The results indicate better performance of existing methods such as MDDM+ on LCDMoir$\acute{e}$. Nevertheless, we argue that LCDMoir$\acute{e}$ dataset is built on simulating the aliasing between CFA and the screen’s LCD subpixel. Compared with smartphone-captured FHDMi with different moir$\acute{e}$ distributions, the moir$\acute{e}$ patterns in LCDMoir$\acute{e}$ are much more uniform and their cropped moir$\acute{e}$ patches are of similar complexity. This explains the relatively poor performance of our approach on LCDMoir$\acute{e}$ dataset. Please note that our approach even improves the performance of baseline models with less computation burden on FHDMi and achieves superior performance in comparison with SOTA networks. Given our motivation for the practical deployment of image demoir$\acute{e}$ing in real cases, we believe the efficacy of the proposed method is still affirmative. We have added the above discussion in Sec.4.3 of our revised manuscript.
>
> **Q3**: The MBCNN method has higher quantitative performance numbers for the LCDMOIRE dataset [2]. For example, it achieves a PSNR of 45.08 in table 6 of that paper. What is the reason for this performance reduction?
>
> **A3**: Sorry for the confusion here. To explain, we actually re-implement the conference version of MBCNN[1] (44.04 dB PSNR, Tensorflow) using the PyTorch framework and achieve a PSNR of 43.95 dB and our pre-trained baseline is available in the supplementary material. Our revised paper has stated this.
>
> [1] Image demoir$\acute{e}$ing with learnable bandpass filters. In Proceedings of the IEEE/CVF Conference on Computer Vision and Pattern Recognition.
>
> **Q4**: A few clarifications on the training process could improve the paper.
>
> **A4**: We appreciate this professional suggestion. In our revised manuscript, we have added more detailed description of the supernet training procedure in Sec.4.1.4.
>
> Thank you again for your time and effort put into our paper. Hopefully, our responses can address all your concerns, such that the score can be kindly reconsidered.

---

> ### Author Response · Authors · 2022-11-17
> **Reminder**
>
> Dear reviewer 9cpB,
>
> Thanks again for your valuable time and insightful comments. As the deadline for the Author/Reviewer discussion is approaching, it would be nice of you to let us know whether our answers have solved your concerns so that we can better improve our work. We are happy to provide any additional clarifications that you may need.
>
> Best regards!
>
> Paper 254 Authors

---

> > ### Comment · Reviewer_9cpB · 2022-11-17
> > **Response to authors after revision**
> >
> > I thank the authors for revising the paper and reflecting on most of the issues which I had pointed out.
> > After reading the revised manuscript and the points mentioned by other reviewers, I still think the fact that the proposed method fails on the LCDMoire dataset is an issue that should be further diagnosed, discussed, and possibly resolved before the publication of this paper. This is considering that the MBCNN method among others, has managed to perform well on both datasets. The underlying reason could be related to the issues the authors pointed to such as being relatively more uniform in terms of moire complexity. But it could also be related to changes in color distribution introduced in LCDMoire, the frequency distribution of the introduced moire, or any other reason. Modifications to the proposed method such as being more adaptive to the distribution of color and moire in the given image instead of mainly focusing on parameters reduction could result in better performance in that dataset. Given that the main ideas in the proposed method are relatively straightforward, such aspects of the study should not be left for future works in my opinion. At the same time, I respect the enthusiasm of two other reviewers, and also considering the efforts and clarifications made by the authors so far, I increase the score to 6.
> >
> >
> > Minor Points:
> >
> > I am still not sure of the training time comparison between the original MBCNN and DMCNN methods versus their DDA variants. I could only see the authors pointing to the number of epochs not for example the training time difference when using the modified loss of Supernet compared to the original ones. This could be a major deciding factor for users of the proposed method.
> >
> > The method Islab-zju mentioned in my main review from the original AIM19 paper is not included in Tab.7. Given that it has outperformed all the mentioned methods, I think it should be included.
> >
> > While this is not affecting my score, the writing could be further improved if some form of grammar check is used. For example, the usage of articles, prepositions, and connecting words could be improved. Bellow, I mention a few issues visible to my eyes:
> >
> > - First paragraph of page 2: "... bring about side effect, ..." ==> "... bring about side effects, ..."
> > - Added text in the middle of page 4: "... colorfuness metric." ==> "... colorfulness metric."
> > - Before section 3.3: "...  computation costs , ..." ==> "...  computation costs, ..."
> > - At the end of section 4.1.4: "All experiments are runned ..." ==> "All experiments are run ..."
> > - Section 4.2.2: "...  while larger width significant outperforms ..." ==> "...  while larger width significantly outperforms ..."

---

> > > ### Author Response · Authors · 2022-11-18
> > > **Thank you for the feedback**
> > >
> > > Thank you for increasing the score. We truly appreciate your effort in helping us to strengthen the paper and your support for our work. Below we address your further comments and suggestions:
> > >
> > > **Q1**: I still think the fact that the proposed method fails on the LCDMoir$\acute{e}$ dataset is an issue that should be further diagnosed, discussed, and possibly resolved.
> > >
> > > **A1**:  With no offense, we would like to emphasize again that the focus of this paper is image demoir$\acute{e}$ing on mobile devices and our motivation comes from the non-uniform distribution of moire patterns in smartphone-captured moire images. Thus, we believe the superior performance on the smartphone-captured FHDMi has already well demonstrated the efficacy of DDA given its original intention. Improving performance on the stimulated LCDMoir$\acute{e}$ requires greater efforts such as modifications on network structure and training paradigm, *etc*, leading to a different methodology. It is hard to finish it within this very limited rebuttal period and does not fit the purpose of this paper. We promise an improved version built upon this paper will be our major concentration after ICLR 2023. Besides, we sincerely appreciate your professional suggestions for the adaptation of DDA to LCDMoir$\acute{e}$, which opens a nice trail for our future work.
> > >
> > > **Q2**: The training time comparison between the original MBCNN and DMCNN methods versus their DDA variants.
> > >
> > > **A2**: Thanks for this suggestion. The training time comparison is now provided in Appendix A.3 of the revised paper.
> > >
> > > **Q3**: The method Islab-zju mentioned in my main review from the original AIM19 paper is not included in Tab.7.
> > >
> > > **A3**: With full respect, the method Islab-zju is just right MBCNN. Please kindly see Sec 4.1 of [1]: *“The Islab-zju team proposed a CNN Based learnable Multiscale Bandpass Filter method, dubbed MBCNN.”*
> > >
> > > [1] Yuan S, Timofte R, Slabaugh G, Leonardis A, Zheng B, Ye X, Tian X, Chen Y, Cheng X, Fu Z, Yang J. Aim 2019 challenge on image demoireing: Methods and results. In International Conference on Computer Vision Workshop (ICCVW) 2019.
> > >
> > > **Q4**: Writing issues.
> > >
> > > **A4**: We really appreciate your efforts for reviewing our manuscript in a very responsible manner. All the mentioned writing issues have been revised in our latest revised paper. Also, the authors have again made great efforts to check the potential errors.

---

### Official Review · Reviewer_Y5JK · 2022-10-25

**Confidence:** 4
**Correctness:** 4
**Technical Novelty And Significance:** 3
**Empirical Novelty And Significance:** 3
**Recommendation:** 8

**Clarity, Quality, Novelty And Reproducibility:**

- The idea is well presented and clearly written
- Code was provided



**Strength And Weaknesses:**

Strengths:
- The frame work for identifying different areas of image to remove the patterns (referred to as DDA) is quite novel
- The ablation study shows that instead of only using frequency or color information, a combined metric shows better performance

I wonder if the image can be segmented using a segmentation network to intelligently divide it into patches?


**Summary Of The Paper:**

The authors present a method of demoireing an image in real time on a mobile device in this body of work. They do this by dynamically selecting areas (patches inside image) by varying complexity and apply different networks to remove the moire pattern.

**Summary Of The Review:**

This paper proposes an innovative method to deal with moire patterns from images. The image is first subdivided and then a color and frequency metric (moire prior) estimates the complexity of the pattern and an appropriate network is applied to remove the moire pattern. The network complexity (supernet) is studied and it's effectiveness is shown in the ablation study.

---

> ### Author Response · Authors · 2022-11-11
> **Reply to Reviewer Y5JK**
>
> We sincerely appreciate your positive and motivating comments. Please kindly see our response to your comment below.
>
> **Q1**: I wonder if the image can be segmented using a segmentation network to intelligently divide it into patches?
>
> **A1**: Thanks for raising this interesting point. The answer is yes if simply considering the performance and we had formulated the same thought when proceeding this work. Unfortunately, directly adopting a segmentation network brings considerable computation costs, which violates our intention for accelerating demoir$\acute{e}$ing networks. Perhaps a more advisable direction is to devise a light-weight and specialized-designed segmentation network to identify the moire complexity, which requires great efforts and will be our future focus.

---

### Author Response · Authors · 2022-11-11
**Summary and general reply to the reviewers**

We thank all the reviewers for their valuable feedback and great efforts, which greatly help us improve the quality of this paper. We have put in a major effort to address all their comments, questions, and concerns. All major modifications in the attached pdf file have been highlighted in blue in order to ease the reading. Note that all the code and models are released in the supplementary material. We first summarize the major changes in our updated version before diving into the detailed point-by-point responses to all the comments:

- Comparison results between our proposed DDA with state-of-the-art networks on LCDMoir$\acute{e}$ dataset.


- Clarifications for the proposed moir$\acute{e}$ prior including the colorfulness and high-frequency metrics.


- Comparison results of one color metric (CIE DeltaE 2000).


- More visualization results for the sorted results of colorful image patches given by our moir$\acute{e}$ prior.


- Phrasing other clarifications requested by reviewers.

---

### Decision · Program_Chairs · 2023-01-20

**Decision:**

Accept: poster

**Justification For Why Not Higher Score:**

(1) The method is simple and efficient, but its generalizability is not fully discussed.
(2) Although the idea is interesting, the application itself (i.e., demoireing) is not an attractive topic in ICLR.

**Justification For Why Not Lower Score:**

As aforementioned, the methodology is interesting and efficient, and the engineering part of this work is solid in my opinion.

**Metareview: Summary, Strengths And Weaknesses:**

In this paper, the authors proposed an efficient image demoireing method that is friendly to mobile devices. In particular, a dynamic mechanism is applied to allocate computation costs adaptively based on the complexity of image patches and the strength of moire patterns.

Strengths:
(1) The idea is simple but efficient, and the engineering implementation is non-trivial.
(2) Experimental results are convincing.

Weaknesses:
(1) The authors should further polish this paper and fix typos.
(2) The authors should add more analytic content and discuss the potential of the proposed method to other learning tasks, which will helps to attract more researchers in this community.

**Note From Pc:**

if the above contains the word "oral" or "spotlight" please see: "oral" presentation means -> notable-top-5% and "spotlight" means -> notable-top-25%. As stated in our emails, we are disassociating presentation type from AC recommendations

**Summary Of Ac-Reviewer Meeting:**

In the rebuttal phase, the reviewers added some comments on comparison experiments and pointed out the typos in the paper. The authors resolved these concerns accordingly. Three of 4 reviewers are satisfied with the authors' reply, and the final scores are 8, 8, 6, 5.
The AC treated the scores evenly and tended to accept this submission.

After the rebuttal phase, AC and reviewers are discussed virtually through messages (the virtual meeting was not set because the time zones of the reviewers and AC are too diverse). The reviewers believe that the paper is a good application-driven paper with a solid engineering part. Although the proposed method is designed for demoireing, the principle of the proposed method is generalized and applicable for other learning tasks.